# Cauda Equina Syndrome Core Outcome Set (CESCOS): An international patient and healthcare professional consensus for research studies

**Nisaharan Srikandarajah**[1]*, **Adam Noble**[2], **Simon Clark**[3], **Martin Wilby**[3], **Brian J. C. Freeman**[4], **Michael G. Fehlings**[5], **Paula R. Williamson**[6], **Tony Marson**[1]

1 Institute of Translational Medicine, University of Liverpool, Liverpool, Merseyside, United Kingdom,
2 Department of Health Services Research, Institute of Population Health Sciences, University of Liverpool, Liverpool, Merseyside, United Kingdom, 3 Department of Spinal Surgery, The Walton Centre NHS Foundation Trust, Liverpool, Merseyside, United Kingdom, 4 Department of Spinal Surgery, Royal Adelaide Hospital, University of Adelaide, Adelaide, Australia, 5 Division of Neurosurgery and Spine Program, Toronto Western Hospital, University Health Network, University of Toronto, Toronto, Ontario, Canada, 6 MRC North West Hub for Trials Methodology Research, Institute of Translational Medicine, University of Liverpool, Liverpool, Merseyside, United Kingdom

* nishsri09@gmail.com

**Data Availability Statement:** All relevant data are within the manuscript and its Supporting Information files.

## Abstract

### Background

Cauda Equina Syndrome (CES) is an emergency condition that requires acute intervention and can lead to permanent neurological deficit in working age adults. A Core Outcome Set (COS) is the minimum set of outcomes that should be reported by a research study within a specific disease area. There is significant heterogeneity in outcome reporting for CES, which does not allow data synthesis between studies. The hypothesis is that a COS for CES can be developed for future research studies using patients and healthcare professionals (HCPs) as key stakeholders.

### Methods and findings

Qualitative semi-structured interviews with CES patients were audio-recorded, transcribed and analysed using NVivo to identify the outcomes of importance. These were combined with the outcomes obtained from a published systematic literature review of CES patients. The outcomes were grouped into a list of 37, for rating through two rounds of an international Delphi survey according to pre-set criteria. The Delphi survey had an overall response rate of 63% and included 172 participants (104 patients, 68 HCPs) from 14 countries who completed both rounds. Thirteen outcomes reached consensus at the end of the Delphi survey and there was no attrition bias detected. The results were discussed at an international consensus meeting attended by 34 key stakeholders (16 patients and 18 HCPs) from 8 countries. A further three outcomes were agreed to be included. There was no selection bias detected at the consensus meeting. There are 16 outcomes in total in the CESCOS.

**Funding:** Educational grants from The Walton Centre charity funds, Coloplast Limited, and Cauda Equina Foundation were greatly appreciated to enable the consensus meeting. The funders had no role in study design, data collection and analysis, decision to publish, or preparation of the manuscript.

**Competing interests:** Coloplast Limited is a funder of this study. There are no patents, products in development or marketed products to declare. This funding does not alter our adherence to PLOS ONE policies on sharing data and materials.

## Discussion

This is the first study in the literature that has determined the core outcomes in CES using a transparent international consensus process involving healthcare professionals and CES patients as key stakeholders. This COS is recommended as the most important outcomes to be reported in any research study investigating CES outcomes and will allow evidence synthesis in CES.

## Introduction

Cauda Equina Syndrome (CES) is an emergency neurological condition that requires acute intervention[1] and can cause significant neurological deficit including bladder, bowel, sexual dysfunction and lower limb paralysis[2, 3]. The incidence of CES is 2 per 100,000 and is an indication for emergency decompression surgery [4–6]. Inadequate management and poor outcomes in CES may result in a high medico-legal burden[7]. CES is commonly categorised into CES incomplete (CESI) and the more severe presentation of CES complete with urinary retention (CESR)[5]. There is little in the literature regarding long term prognosis[8] and a review of studies evaluating treatments for CES demonstrated heterogeneity in the outcome domains measured [9]. In addition, the outcomes reported in the literature have not been independently validated as important by key stakeholders. A Core Outcome Set (COS) is "an agreed, standardised set of outcomes to be measured and reported, as a minimum, in all trials for that particular health area"[10]. The concept of a COS was developed to standardise outcomes across all relevant trials to allow comparisons of the results of different trials in a given condition [11].

### Objectives

This paper reports the consensus process which was undertaken with key stakeholders (patients and healthcare professionals [HCPs]) to achieve the Cauda Equina Syndrome Core Outcome Set (CESCOS). A systematic literature review and qualitative interviews were conducted to identify a complete list of outcomes. These outcomes populated a two-round Delphi survey, which participants completed and reviewed at a consensus meeting. Key stakeholders identified the most important outcomes but the group did not intend to validate how to measure these outcomes in this study. This study is reported in accordance with the Core Outcome Set-STAndards for Reporting (COS-STAR)[12] guidelines.

### Scope

The health condition included all severities of CES. The population involved are adults with CES over the age of 18 years. The intervention was medical and surgical management of CES and the setting where the COS is to be applied is for any CES related research study.

## Methods

### Protocol/ Registry entry

The CESCOS is officially registered on the Core Outcome Measures in Effectiveness Trials (COMET) database as study 824 (http://www.comet-initiative.org/studies/details/824). Details regarding the methods are described in further detail in the protocol[13].

## Participants

Participants for the CESCOS Delphi survey were recruited from two key stakeholder groups: patients with CES and HCPs who manage CES patients. All were adults aged over 18 years and able to independently complete an online questionnaire in English. Participants were recruited from a database at the local site, through snowball sampling [14] of known contacts and through international and national HCP and patient organisations.

## Information sources

A published systematic literature review (SLR) [9] identified all the outcomes documented in studies since 1990 involving patients who had undergone surgery for CES. The outcomes from the SLR were combined with the outcomes identified from the qualitative interviews to form those initially rated on within the Delphi Survey. These qualitative interviews had been conducted by NS with 22 patients treated at The Walton Centre between 2007 and 2016 for CES. A sampling frame was applied to ensure patients with a range of CES severities (CESI or CESR) and different times since the operation were interviewed. Semi-structured interviews were conducted with a topic guide (**S1 File**) and involved patients' describing their experience of CES in a chronological manner to ascertain the relevant outcomes and the lived experience of the condition. Interviews were audio recorded, transcribed and with the assistance of NVivo (version 10), were coded using an inductive approach to identify outcomes. NS led the analysis process and was supported by AN.

The SLR produced 737 verbatim outcome terms and the qualitative interviews identified 260. The qualitative interviews highlighted 43 verbatim outcome terms not identified by the literature review, which were more concerned with life impact. There was a total of 997 verbatim outcome terms, which was condensed by the study team to 37 outcomes. These were categorised into five higher order categories as per the taxonomy recommended by COMET (Clinical Outcomes, Life Impact, Resources Use, Death and Adverse Events) [15].

The process of reducing the "long list" to a "short list" of outcomes was reviewed by the study team including patient research partners for face validity, understanding and acceptability and modified according to feedback. For example, regarding low back pain there were 53 verbatim outcome terms from the SLR (n = 31) and the qualitative interviews (n = 22) but these were all summarised to one outcome of low back pain. In addition, the terminology and explanations of the outcomes were decided using the language from the patient interviews and refined through a series of cognitive "think aloud" interviews conducted with HCPs and patient representatives [16, 17].

## Consensus process

**Delphi survey.** The "modified" Delphi method [18] was used with outcomes derived from the SLR and interviews. Additional outcomes were suggested in round 1 by participants. Demographic details were collected on the registration page. The Delphi survey was anonymised and only participants who responded to the first round of the Delphi were invited to participate in the second round. Data was collected over a 4-week period for each Delphi round. The setup and running of the survey were managed by using the DelphiManager software [19].

**Consensus meeting.** All participants needed to complete both rounds of the Delphi survey to be eligible to attend the consensus meeting. A sampling frame was used to achieve a varied sample of participants and representation from key stakeholder organisations. The meeting was chaired by a trained non-clinical independent facilitator (SB) not on the study team. Forty participants (20 patients and 20 HCPs) were invited to the consensus meeting: fifteen participants in each group were from the UK and five in each group were from outside the UK.

## Outcome scoring

**Delphi survey.** Participants were asked to rate each outcome using a 9-point Likert scale (7 to 9 indicating critical importance, 4 to 6 representing outcomes that are important but not critical, 1 to 3 are deemed to be of limited importance). All outcomes were retained for voting in the second round and presented with their anonymised first round scores from the patient and HCP groups. Participants could decide to keep or change their original answers on second thoughts. Attrition bias was assessed by comparing the average scores of participants who completed both rounds to the average score of the participants who only completed round one.

**Consensus meeting.** The main discussion at the consensus meeting considered the outcomes with "No consensus" in the Delphi survey (**Table 1**). Participants at the meeting voted on these outcomes anonymously using the TurningPoint system and handsets (Turning Technologies, Youngstown, OH, USA).

## Consensus definition

We have adopted the "70/15" consensus definition in the protocol, which was used successfully in other COS studies [20, 21] for inclusion of an outcome in the COS. However, it was partially revised for "consensus out" due to the study team's experience from other studies where outcomes were rarely voted 1–3 not important and reach criteria for exclusion after the Delphi survey[21]. This revision was done without reference to the identity of the outcomes. As a result, the final definitions of consensus that were used are in **Table 1.** The same criteria were used for the consensus meeting. All outcomes in the "consensus out" or "no consensus" category after voting in the consensus meeting were not included in the COS. Feedback forms were distributed and collected at the end of the meeting.

## Ethics

Research Ethics Committee (REC) and Health Research Authority (HRA) approval was obtained on December 2016 for the qualitative interviews by South Central- Hampshire A Research Ethics Committee (REC reference 16/SC/0587). REC and HRA approval was obtained on March 2018 for the Delphi process and consensus meeting by North West-Greater Manchester Central Research Ethics Committee (REC reference 18/NW/0022).

# Results

## Protocol deviations

As mentioned before, the definition for an outcome not to be included (termed "consensus out" in **Table 1**) was changed for the Delphi survey with agreement from the study team. There were no other deviations from the protocol.

**Table 1. Definitions of consensus for the Delphi survey and consensus meeting.**

| Classification of consensus | Description | Definition |
|---|---|---|
| IN | Consensus that an outcome should be included in the core outcome set | 70% or more participants scoring as 7 to 9 AND <15% participants scoring as 1 to 3 in both the patient and HCP groups |
| OUT | Consensus that an outcome should not be included in the core outcome set | ≤50% of participants scoring as 7 to 9 in both the patient and HCP groups |
| NO CONSENSUS | Uncertainty about importance of an outcome | Anything else |

## Participants

**Delphi survey.** HCP and patient organisations who circulated the Delphi survey amongst their membership are listed in the S1 Table. Round one was completed by 272 participants (189 patients, 83 HCPs). Both rounds were completed by 172 participants. Sixty percent were patients (104) and 40% were HCPs (68). The overall response rate was 63% (55% for patients and 82% for HCPs). The patient (Table 2) and HCP demographics (Table 3) are available below.

**Consensus meeting.** Thirty-four participants attended the consensus meeting (16 patients and 18 HCPs). Twenty-five participants were from the UK and 9 were international. There was international patient and healthcare representation from CES charity organisations and

**Table 2. Demographics of patient Delphi participants who completed both rounds.**

| PATIENTS | n (%) |
|---|---|
| Total | 104 |
| **Gender** | |
| Male | 26 (25) |
| Female | 78 (75) |
| **Age group** | |
| 18–29 | 6 (6) |
| 30–39 | 30 (29) |
| 40–49 | 31 (30) |
| 50–59 | 22 (21) |
| 60–69 | 13 (13) |
| 70+ | 2 (2) |
| **Country of residence** | |
| UK | 54 (52) |
| USA | 40 (38) |
| Ireland | 2 (2) |
| Denmark | 2 (2) |
| Canada | 2 (2) |
| Australia | 2 (2) |
| Brazil | 1 (1) |
| South Africa | 1 (1) |
| **CES diagnosis** | |
| <2 | 36 (35) |
| 2–5 | 27 (26) |
| 5–10 | 23 (22) |
| >10 | 18 (17) |
| **Employment status** | |
| Employed full time | 30 (29) |
| part time | 10 (10) |
| Self employed | 9 (9) |
| Unemployed | 6 (6) |
| Unable to work | 29 (28) |
| Homemaker | 5 (5) |
| Retired | 14 (13) |
| Not answered | 1 (1) |
| **CES Operation** | |
| Yes | 89 (86) |
| No | 15 (14) |

**Table 3. Demographics of HCP Delphi participants who completed both rounds.**

| HCPs | n (%) |
|---|---|
| **Total** | 68 |
| **Gender** | |
| Male | 60 (88) |
| Female | 8 (12) |
| **Occupation** | |
| Neurosurgery | 36 (53) |
| Orthopaedic | 12 (18) |
| Neuro-rehabilitation | 5 (7) |
| Neurologist | 4 (6) |
| Spinal Cord Injury | 4 (6) |
| Spinal nurse | 3 (4) |
| Physiotherapist | 2 (3) |
| Psychologist | 2 (3) |
| **Years of practice (as a consultant/ attending after board certification)** | |
| <2 | 4 (6) |
| 2–5 | 6 (9) |
| 5–10 | 14 (21) |
| 10–20 | 24 (35) |
| 20+ | 19 (28) |
| Not stated | 1 (1) |
| **Country of residence** | |
| UK | 41 (60) |
| Canada | 11 (16) |
| Portugal | 3 (4) |
| Ireland | 2 (3) |
| Germany | 2 (3) |
| Australia | 2 (3) |
| India | 2 (3) |
| Czech Republic | 1 (1) |
| USA | 1 (1) |
| Brazil | 1 (1) |
| New Zealand | 1 (1) |
| Malaysia | 1 (1) |

HCP spine and rehabilitation organisations. The consensus meeting was chaired by a non-clinical researcher (SB) independent to the study team with expertise in core outcome set methodology.

Of the 18 HCPs at the consensus meeting, 10 were surgeons involved in acute CES management and 8 were doctors and allied HCPs involved in the longer-term care and rehabilitation of CES patients. In the patient group, there was an equal spread of patients in the years since diagnosis of CES (<2: 5, $\geq 2 < 5$: 4, $\geq 5 < 10$: 6 and $\geq 10$: 1).

When comparing average round two Delphi scores between participants who attended the consensus meeting (patients mean 7 SD 1: HCPs mean 7 SD 0.7) to those participants who did not attend the consensus meeting (patients mean 7 SD 0.85: HCPs 7 mean SD 1), there was no participation bias.

## Outcomes

**Delphi survey.** The list of the outcomes and agreed terminology with explanations used in the Delphi survey are available in **S2 Table**. There was a total of 37 outcomes. Sixty-five additional outcomes were suggested at the end of round 1 but only one outcome of "pain from abnormal sensation or non-painful stimulus" was deemed appropriate to be included for round 2. The other 64 suggestions were not included as 33 (52%) were not an outcome, 30 (47%) were covered by other outcomes already on the Delphi survey and 1 (1%) suggestion was not due to CES.

**Table 4** shows the percentage of participants who had voted 7 to 9 (critically important) for each outcome at the end of rounds 1 and 2 of the Delphi survey. According to the pre-specified scoring criteria (**Table 1**), 13 outcomes were included as "consensus in," (green), 6 were "consensus out" (blue) and 19 had "no consensus" at the end of both rounds. During the entry of the outcome "pain from abnormal sensation or non-painful stimulus" for rating in round two the "sensation in genitals" outcome was accidently deleted. Both these outcomes achieved "consensus in" in the one round they were rated in so it was agreed by the study team to include them in the list of "consensus in" outcomes.

In green are the outcomes that were included and blue were the outcomes excluded. X denotes that an outcome has not been voted in the round.

There were 499 score changes in total in round 2. Patients made 326 (65%) score changes and 173 (35%) were from HCPs. Most patients made score change based on personal reflection (71%) whereas most HCPs (58%) had made the score changes based on stakeholder feedback. The mean round 1 scores for patients (mean 7 SD 1.02) and HCPs (mean 6 SD 0.87) were not different compared to the participants that completed both rounds for patients (mean 7 SD 0.93) and HCPs (mean 6 SD 0.87). This suggests that there was no attrition bias.

**Consensus meeting.** **Table 5** shows the percentage of participants that voted 1–3, 4–6 and 7–9 for the "No consensus" outcomes in the consensus meeting. Three further outcomes were included in the COS after voting in the consensus meeting; sensation of bladder fullness, low mood and depression and social functioning.

The outcome that was re-voted on in the consensus meeting was low mood and depression. The HCPs incorrectly assumed that since the outcome of global quality of life was in the COS then low mood and depression would be automatically included in this outcome. The facilitator highlighted that all quality of life measurements would not measure the same outcomes and if participants wanted an outcome related to quality of life to be included they had to vote it in. After adequate discussion, a re-vote was agreed by the study team, which resulted in the outcome of low mood and depression being included. The outcome of death was deemed to be already covered by the outcome of complications and the study team agreed to include this in the definition of complications hence it was not voted on.

Foot drop, back pain and need for further intervention were voted as critically important by patients but not by HCPs. Low back pain was not voted critically important by HCPs as they felt it was due to several different causes so to ascribe it to CES would be incorrect. With regards to foot drop this was not included as HCPs and some patients felt the outcome of mobility and walking would encompass the effects experienced by foot drop. HCPs felt the need for further intervention was already included within the outcome of complications. They also felt further procedures related to the management of CES would say little regarding the effectiveness of the initial intervention for CES.

The "no consensus" outcomes which were critically important by <70% of participants from both stakeholder groups in the Delphi survey were agreed by the consensus meeting participants to not be voted on and to accept the results of the Delphi; Sexual desire, constipation, sensation in the legs, urinary urgency and abdominal pain. Fatigue although in this category

**Table 4. Percentage of patients and HCPs scoring 7–9 (critical) for an outcome in rounds 1 and 2.**

| Outcome | Patients (n = 189) R1 | HCPs (n = 83) R1 | Patients (n = 104) R2 | HCPs (n = 68) R2 |
|---|---|---|---|---|
| Urinary retention | 74 | 93 | 80 | 97 |
| Sensation of bladder fullness | 69 | 61 | 74 | 63 |
| Incontinence of Urine | 76 | 91 | 84 | 100 |
| Urinary urgency | 57 | 30 | 55 | 36 |
| Urinary frequency | 48 | 27 | 43 | 31 |
| Constipation | 67 | 25 | 66 | 31 |
| Faecal Incontinence | 80 | 94 | 89 | 99 |
| Abdominal distention | 49 | 18 | 42 | 12 |
| Abdominal pain | 54 | 23 | 52 | 24 |
| Anal tone | 63 | 57 | 76 | 69 |
| Physical ability to have sexual intercourse | 80 | 81 | 84 | 92 |
| Leg muscle strength | 71 | 67 | 80 | 72 |
| Foot drop | 64 | 60 | 76 | 60 |
| Reflexes | 51 | 11 | 44 | 3 |
| Sensation in leg(s) | 66 | 40 | 63 | 32 |
| Pain from abnormal sensation or non-painful stimulus | X | X | 85 | 81 |
| Genital Sensation | 82 | 72 | X | X |
| Perineal sensation | 74 | 65 | 75 | 73 |
| Lower back pain | 83 | 29 | 83 | 35 |
| Pain in leg and/or feet | 82 | 48 | 83 | 53 |
| Back stiffness | 53 | 10 | 47 | 6 |
| Leg stiffness | 48 | 11 | 48 | 7 |
| Fatigue | 56 | 16 | 56 | 15 |
| Non-specific pain | 48 | 8 | 36 | 6 |
| Global Quality of Life | 85 | 80 | 90 | 75 |
| Occupation/ Role functioning | 72 | 81 | 85 | 88 |
| Social functioning | 62 | 70 | 66 | 73 |
| Ability to do Daily activities (Physical functioning) | 81 | 80 | 89 | 90 |
| Mobility and Walking (Physical functioning) | 86 | 82 | 91 | 88 |
| Difficulty with body posture (Physical functioning) | 60 | 52 | 70 | 50 |
| Sexual desire (Emotional functioning) | 64 | 64 | 65 | 65 |
| Anxiety (Emotional functioning) | 69 | 51 | 74 | 49 |
| Isolation (Emotional functioning) | 72 | 56 | 74 | 59 |
| Low Mood and Depression (Emotional functioning) | 75 | 58 | 78 | 63 |
| Hospital resources | 74 | 46 | 83 | 51 |
| Need for further intervention | 84 | 51 | 89 | 53 |
| Death | 54 | 59 | 66 | 72 |
| Complications | 78 | 65 | 82 | 72 |

was requested by the patient stakeholder group to be voted on again. Other outcomes already included in the COS were contributory to fatigue such as mobility and walking, ability to do daily activities and leg muscle strength and this was cited as a reason by a HCP and patient as not choosing it critically important. It did not reach the criteria for inclusion in the COS.

The feedback for the consensus meeting was completed by 13 out of 16 patients (81%) and 16 out of 18 HCPs (89%). From the completed responses, 100% agreed that the meeting produced a fair result and they were comfortable communicating their views.

**Table 5. Percentages of patients and HCPs who voted 1–3 (not important), 4–6 (important but not critical), 7–9 (critical) for the "no consensus" outcomes at the consensus meeting.**

| Outcome | Patient (n = 16) (%) | | | HCP (n = 18) (%) | | |
|---|---|---|---|---|---|---|
| | 1–3 | 4–6 | 7–9 | 1–3 | 4–6 | 7–9 |
| Anal tone | 19 | 62 | 19 | 29 | 62 | 12 |
| Sensation of bladder fullness | 0 | 12 | 87 | 0 | 23 | 78 |
| Foot drop | 0 | 12 | 88 | 0 | 50 | 50 |
| Pain in leg or feet | 0 | 44 | 56 | 6 | 61 | 34 |
| Back Pain | 0 | 12 | 88 | 12 | 29 | 62 |
| Low mood and depression | 0 | 0 | 100 | 0 | 17 | 83 |
| Social functioning | 0 | 12 | 88 | 0 | 28 | 72 |
| Isolation | 0 | 69 | 31 | 0 | 72 | 28 |
| Anxiety | 0 | 31 | 69 | 0 | 50 | 50 |
| Difficulty of body posture | 0 | 50 | 51 | 0 | 83 | 17 |
| Need for further intervention | 0 | 19 | 82 | 0 | 44 | 56 |
| Hospital resources | 6 | 82 | 13 | 17 | 72 | 11 |
| Fatigue | 0 | 33 | 67 | 0 | 78 | 22 |

## The core outcome set

The final COS is listed in **Table 6**. There are 16 outcomes in total categorised under autonomic function, non-autonomic function and quality of life.

## Overview

The SLR (737) and qualitative interviews (260) identified 997 verbatim outcome terms. This was then prioritised through a Delphi survey with 37 outcomes in total. An additional outcome was added in the second round creating 38 outcomes. At the end of the Delphi survey 13 outcomes were included in the COS. This was agreed at the consensus meeting and 3 extra outcomes were included to the COS creating a total of 16 outcomes (**Fig 1**).

**Table 6. The 16 outcomes that constitute the Cauda Equina Syndrome core outcome set.**

| CES Core Outcome Set | | |
|---|---|---|
| Autonomic function | Bladder function | Incontinence of Urine |
| | | Urinary retention |
| | | Sensation of bladder fullness |
| | Bowel function | Faecal incontinence |
| | Sexual function | Physical ability to have sexual intercourse |
| | Sensation | Perineal sensation |
| | | Sensation in genitals |
| Non-autonomic function | Power | Leg muscle strength |
| | Pain | Pain due to abnormal sensation or non-painful stimulus |
| | Adverse Events | Complications (including death) |
| Quality of life | | Global quality of life |
| | | Occupational role functioning |
| | | Social functioning |
| | | Ability to do daily activities |
| | | Mobility and walking |
| | | Low Mood and depression |

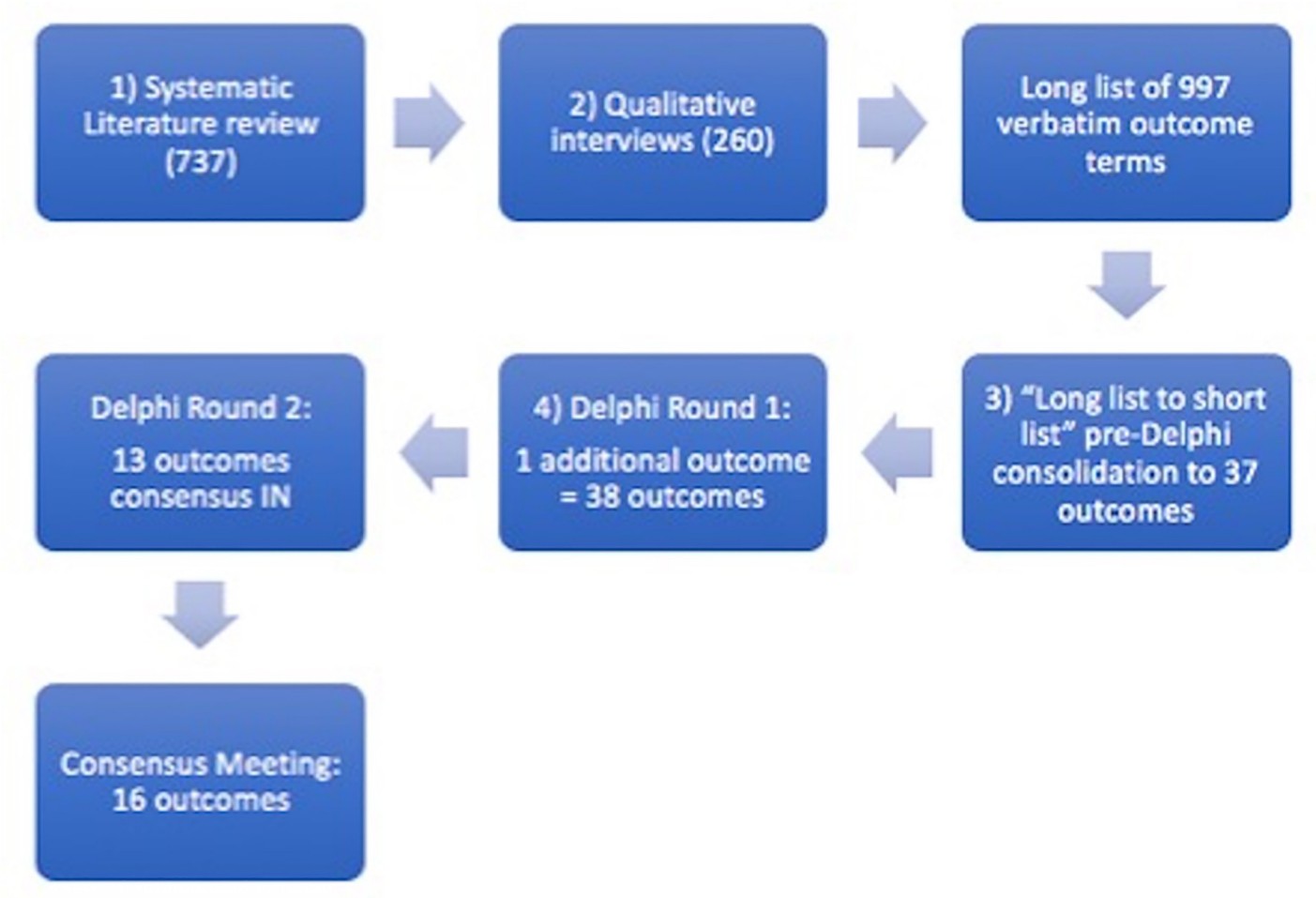

**Fig 1. Overview of core outcome set development and the final Cauda Equina Syndrome core outcome set.**

## Discussion

This is the first study in the literature that has determined the Core Outcomes Set (COS) for CES. It was registered on the COMET database and a transparent process has been used involving an international Delphi survey and consensus meeting to decide the COS. Each outcome included has been scored and agreed as critical by at least 70% of patients and 70% of HCPs. This COS is recommended for use in any research study investigating CES outcomes as they have been verified as important by key stakeholders. This will allow evidence synthesis in CES.

It is noted that the numbers recruited in a Delphi survey can potentially be small if the condition is rare[19]. A review of COS studies from the COMET database revealed that 22% had recruited patients from 5 or more countries [22]. Inclusion of patients from multiple countries is deemed more difficult than HCPs [19]. The CESCOS study recruited 172 participants for both rounds and involved patients from 8 countries and HCPs from 12 countries. Participants from the UK made up 55% and 45% were international of who most were patients. CES is a rare condition so this was deemed to be a satisfactory response.

For the CESCOS Delphi, HCPs (82%) had a better response rate compared to the patients (55%) in round 2. This may be a reflection that most HCPs were recruited from professional

organisations and patients were recruited openly from social media[21]. The importance of completing both rounds of the Delphi may not have been emphasised enough through social media. However, there was no attrition bias detected in the results of the CESCOS study.

Most HCPs taking part in the Delphi were of a surgical background and 63% had 10 years or more experience after board certification as a consultant/ attending or the equivalent. This is reflective of current CES management and research as it is managed as an acute condition requiring emergency intervention in most cases[5, 6, 23]. Fifty percent patients were in the age group of 30 to 49 and 52% of patients were not in employment or retired. This reflects that CES adversely affects a working age population. Eighty-nine percent of patients had an operation for CES and this is consistent with the main aetiology for CES being a compressive pathology which requires surgical decompression[4, 5]. As mentioned in the results, over half of the HCPs attending the consensus meeting were involved in acute management and the rest involved in longer term care and rehabilitation of CES patients, which would be reflective of a group of HCPs that manage CES patients in the short and long term from diagnosis. There was an equal spread of patients in the years since CES diagnosis, which would have also facilitated prioritisation of short and long term outcomes.

In Round 1, a higher proportion of HCPs scored autonomic related outcomes (urinary retention, incontinence of urine, faecal incontinence and physical ability to have sexual intercourse) as critically important compared to patients. Outcomes scored higher by patients in Round 1 included genital sensation and life impact outcomes such as global quality of life, ability to do daily activities and mobility and walking. This agrees with the literature where HCPs prioritise clinical outcomes compared to those related to life impact, which patients find important. This was also reflected at an earlier stage when the verbatim outcome terms, which were mentioned in the qualitative interviews and not in the SLR were mainly related to life impact. There is evidence which suggests that patients tend to rate many or all outcome domains as important in prioritisation exercises so HCP views would dominate as the outcome domains they do not deem important will not be included in the final COS [24]. This was observed for ten outcomes in the CES Delphi survey where ≥70% of patients voted them critical but HCPs had not therefore excluding them from the core outcome set at this stage (**Table 4**). These outcomes were sensation of bladder fullness, anal tone, foot drop, low back pain, leg pain, difficulty with body posture, anxiety, isolation, low mood and depression and hospital resources. The outcome of anal tone, which has been measured in CES research studies[9, 25] is used as a proxy for faecal incontinence. However, anal tone was not voted into the COS but faecal incontinence was, which highlights the importance of not just measuring what clinicians believe is important.

Multiple group feedback between rounds has been shown to improve consensus between stakeholder groups[26]. The CESCOS used this feedback method and found most HCPs (58.4%) and some patients (27.6%) made score changes based on the feedback from the stakeholder groups. This led to consensus on 13 outcomes to be included and 6 to be excluded at the end of the Delphi survey. No participation bias was seen with the participants who attended the consensus meeting.

A prospective study of long term outcomes after surgery for 46 CES patients had a mean follow up of 43 months[27]. Validated questionnaires and unvalidated semi structured interviews were used to assess long term outcomes of bladder, bowel, sexual and physical function. Not all the outcomes in the CESCOS have been measured. For example, perineal sensation, sensation in genitals, leg muscle strength, pain due to abnormal sensation or non-painful stimulus and complications from the operation, were not measured. For the outcomes that were measured, there has not been a transparent consensus process[28] regarding the choice of these outcome measurement instruments. In Table 5 of a systematic literature review[9], it was

shown that between studies, there is a lack of uniformity in the assessments used for the outcomes in CES, which makes it difficult to synthesise the results for meaningful analysis. The CESCOS highlights the outcomes for which this process must be undertaken in a transparent and methodologically sound manner.

There is little research into the uptake of core outcome sets in comparison to randomised trials and systematic reviews as there are a relatively smaller number of them[19]. A review of the rheumatoid arthritis COS established in 1994 showed that 81% of trials between 2002 and 2016 were reporting it[29]. Implementation is an important aspect to help promote uptake and to aid this the CESCOS is registered on the COMET database, is published and will be presented at meetings and relevant HCP, patient and research funding bodies will be informed.

In the qualitative interviews, Delphi survey and consensus meeting for the CESCOS study, patients were keen and willing to be involved. This may be a reflection that it is a rare syndrome so any attention or further research for the condition is actively engaged and welcomed by them suggesting the burden of data collection is not an issue with this patient group. A study reviewing the long-term outcomes after spinal surgery for CES showed that there was a real need for HCPs to spend sufficient time discussing the difficult issues and delivering prognostic information to patients regarding their outcomes[27]. This highlights the importance of recording and reporting the CESCOS outcomes for future research studies. Currently, a multi-centre prospective observational cohort study in CES is using the CESCOS[30] for its follow up data collection. The pragmatic difficulties of data measurement will be explored here. The COS should be reviewed in the future to see if any outcomes need to be added or subtracted [19]. The aim is to do this in five years to analyse uptake in CES research studies.

## Strengths and limitations

A varied sample was obtained for the qualitative interviews using a sampling frame, which identified outcomes important to patients. The Delphi survey recruited participants from 14 countries and the consensus meeting recruited participants from 8 countries, which is significant considering CES is a rare condition. The consensus process successfully involved both patient and HCPs in the prioritisation of outcomes and agreement over the COS.

The study was only conducted in the English language due to time and budget resource limitation. During the Delphi survey, details of how patients presented with CES were not collated as it would not have been possible to verify these details with the respective medical notes within the time limitations of the study.

## Conclusion

We have determined 16 outcomes that are critical to key stakeholders (**Table 6**). In the medical literature, there is a focus on the bladder dysfunction and clinical sequelae of CES [9]. There is little emphasis on outcomes related to life impact. This COS has highlighted the importance of all these outcomes to be assessed as the "minimum standard." To ensure consistency in measurement and reporting of these outcomes the next stage will involve gaining consensus around standardised definitions and recommended measurement instruments for each outcome in the COS following the COSMIN-COMET guidelines [28].

## Supporting information

**S1 File. Topic guide for the qualitative interviews.**
(DOCX)

**S1 Table. Patient and HCP organisations that circulated the Delphi.**
(DOCX)

**S2 Table. List of outcomes with their associated plain language and clinical explanations used for the Delphi.**
(DOCX)

## Acknowledgments

CES consensus meeting collaborators (Healthcare professionals and Patients) who attended and participated in the consensus meeting include:

**Healthcare professionals:** Charles Davis, Fadel Derry, Canisius Dzapasi, Michael G Fehlings, Paulo Roberto Franceschini, Brian J C Freeman, Elizabeth Heaps, Radek Kaiser, Rafid Al-Mahfoudh, Siva Nair, Tim Pigott, Zaid Sarsam, Helen Smith, Bakulesh Soni, Marco Teli, Christos Tolias, Laura Whitfiled, S L Yadav

**Patients:** Paul Chapman, Ann Derbyshire, Natalie Falconer, Darren Hayden, Barry Kirby, Patricia Knight, Billy Maskell, Pat Mason, Mark Paine, Julie Pearson, Amanda Proctor, Claire Rawlings, Steven Smith, Anja Tribler, Claire Thornber, John Robert Whitehead

Special thanks to Claire Thornber and Steven Smith for their role as patient representatives on the study team. Thanks to Dave Watling, the research manager at The Walton Centre who helped with the costings and arrangement of the consensus meeting. We thank Richard Crew for support with DelphiManager, Heather Bagley for guidance on patient involvement at the consensus meeting and Sara Brookes for chairing the consensus meeting. Educational grants from The Walton Centre charity funds, Coloplast Ltd and Cauda Equina Foundation were greatly appreciated to enable the consensus meeting. We are grateful to the following patient and HCP organisations who distributed the Delphi survey; Cauda Equina Syndrome Association, Cauda Equina Foundation, Spinal Injuries Association, Brain and Spine Foundation, Society of British Neurological Surgeons, Eurospine, Canadian Spine Society, International spinal cord society, Spine society Australia, World federation of neuro-rehabilitation and the British Society of rehabilitation medicine.

## Author Contributions

**Conceptualization:** Nisaharan Srikandarajah, Simon Clark, Martin Wilby, Paula R. Williamson, Tony Marson.

**Data curation:** Nisaharan Srikandarajah, Adam Noble.

**Formal analysis:** Nisaharan Srikandarajah, Adam Noble, Simon Clark, Martin Wilby, Brian J. C. Freeman, Michael G. Fehlings, Paula R. Williamson, Tony Marson.

**Investigation:** Nisaharan Srikandarajah.

**Methodology:** Nisaharan Srikandarajah, Adam Noble, Simon Clark, Martin Wilby, Brian J. C. Freeman, Michael G. Fehlings, Paula R. Williamson, Tony Marson.

**Project administration:** Nisaharan Srikandarajah, Tony Marson.

**Resources:** Nisaharan Srikandarajah.

**Software:** Paula R. Williamson.

**Supervision:** Adam Noble, Simon Clark, Martin Wilby, Michael G. Fehlings, Paula R. Williamson, Tony Marson.

**Validation:** Nisaharan Srikandarajah, Adam Noble, Martin Wilby, Brian J. C. Freeman, Michael G. Fehlings, Paula R. Williamson, Tony Marson.

**Visualization:** Paula R. Williamson, Tony Marson.

**Writing – original draft:** Nisaharan Srikandarajah.

**Writing – review & editing:** Nisaharan Srikandarajah, Adam Noble, Simon Clark, Martin Wilby, Brian J. C. Freeman, Michael G. Fehlings, Paula R. Williamson, Tony Marson.

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
