## [Decision Letter · Decision Letter 0]

18 Sep 2019

PONE-D-19-18530

Cauda Equina Syndrome Core Outcome Set (CESCOS): An international patient and healthcare professional consensus for research studies

PLOS ONE

Dear Mr Srikandarajah,

Thank you for submitting your manuscript to PLOS ONE. After careful consideration, we feel that it has merit but does not fully meet PLOS ONE’s publication criteria as it currently stands. Therefore, we invite you to submit a revised version of the manuscript that addresses the points raised during the review process.

We would appreciate receiving your revised manuscript by Nov 02 2019 11:59PM. To enhance the reproducibility of your results, we recommend that if applicable you deposit your laboratory protocols in protocols.io, where a protocol can be assigned its own identifier (DOI) such that it can be cited independently in the future. For instructions see: http://journals.plos.org/plosone/s/submission-guidelines#loc-laboratory-protocols

We look forward to receiving your revised manuscript.

Kind regards,

Rodrigo Romao

Academic Editor

PLOS ONE

Journal Requirements:

2. Please include a copy of the interview guide used in the study, in both the original language and English, as Supporting Information, or include a citation if it has been published previously.

3. One of the noted authors is a group or consortium [CES Consensus Group]. In addition to naming the author group, please list the individual authors and affiliations within this group in the acknowledgments section of your manuscript. Please also indicate clearly a lead author for this group along with a contact email address.

Additional Editor Comments (if provided):

Reviewers' comments:

Reviewer's Responses to Questions

**Comments to the Author**

1. Is the manuscript technically sound, and do the data support the conclusions?

Reviewer #1: Yes

Reviewer #2: Yes

2. Has the statistical analysis been performed appropriately and rigorously? 

Reviewer #1: Yes

Reviewer #2: Yes

3. Have the authors made all data underlying the findings in their manuscript fully available?

Reviewer #1: Yes

Reviewer #2: Yes

4. Is the manuscript presented in an intelligible fashion and written in standard English?

Reviewer #1: Yes

Reviewer #2: Yes

5. Review Comments to the Author

Reviewer #1: This study was clearly described in terms of relevance, methodology and results. It appears to be very novel and highly relevant for future research on patients with CES. I found it interesting that bladder dysfunction scored so highly with regards to importance as a COS.

Reviewer #2: Thank you for asking me to review this submitted manuscript.

The authors report an international patient and healthcare professional consensus and propose a Cauda Equina Syndrome Core Outcome Set (CESCOS).

Some comments:

1. Overall, the authors seem to be appropriately building on their previous work [(Spine (Phila Pa 1976). 2018 Sep 1;43(17):E1005-E1013. doi: 10.1097/BRS.0000000000002605)], where significant heterogeneity in outcomes reported for studies after surgery for CES patients indicated a need for the development of a core outcome set.

2. Abstract: reasonable depiction of the manuscript.

3. Introduction: fine

4. Methods: described logically and comprehensively.

5. Results:

- In "Protocol Deviations", the first sentence is difficult to understand due to the word "out". Please rephrase.

- The first two subheadings ('Protocol deviations' and 'The List of Outcomes') seem to be more appropriately placed in the Methods section.

- please reserve the Results section for Results alone.

- In reporting the "years of practice" amongst HCP participants, do the authors include years in training?

- Can the authors clarify how many respondents were trainees vs specialists?

- Their later comment in the discussion that "63% had 10 years or more experience" is noted, but it remains unclear whether the 10 is post-residency training.

6. Discussion:

- appropriate discussion of the consensus group composition.

- the authors have omitted to cite a recent study (perhaps because this was published after the present study's submission) which seemingly reported the largest cohort of patients with CES investigated for long-term outcomes using validated questionnaires. Please discuss the findings and choice of questionnaires used, and how your COS might improve this. [Hazelwood JE et al. An assessment of patient-reported long-term outcomes following surgery for cauda equina syndrome. Acta Neurochir (Wien). 2019 Sep;161(9):1887-1894. doi: 10.1007/s00701-019-03973-7. Epub 2019 Jul 1.]

- Limitations: fair discussion.

- can the authors discuss the potential uptake of their recommended COS, incl the length of time required to obtain the COS data and any pragmatic difficulties.

7. Conclusion:

- 16 outcomes that are critical to key stakeholders have been identified. How

8. This constitutes a piece of hard work and the authors are congratulated in its design and execution.

Looking forward to a quick return addressing the points made.

6. PLOS authors have the option to publish the peer review history of their article (what does this mean?). If published, this will include your full peer review and any attached files.

Reviewer #1: Yes: Ashley Cox

Reviewer #2: No

---

## [Author Response · Author response to Decision Letter 0]

11 Nov 2019

Response to Reviewers

PONE-D-19-18530

Cauda Equina Syndrome Core Outcome Set (CESCOS): An international patient and healthcare professional consensus for research studies

PLOS ONE

All comments from the journal and reviewers have been highlighted in bold and addressed below in normal script.

Journal Requirements:

This has been done.

2. Please include a copy of the interview guide used in the study, in both the original language and English, as Supporting Information, or include a citation if it has been published previously.

This has been done: 

“Semi-structured interviews were conducted with a topic guide (S1 File) and involved patients’ describing their experience of CES in a chronological manner to ascertain the relevant outcomes and the lived experience of the condition.”

3. One of the noted authors is a group or consortium [CES Consensus Group]. In addition to naming the author group, please list the individual authors and affiliations within this group in the acknowledgments section of your manuscript. Please also indicate clearly a lead author for this group along with a contact email address.

This has been deleted from the author byline. In the acknowledgements, I have included patient and healthcare collaborators for the consensus meeting.

Addressing reviewer comments

Reviewer #1: This study was clearly described in terms of relevance, methodology and results. It appears to be very novel and highly relevant for future research on patients with CES. I found it interesting that bladder dysfunction scored so highly with regards to importance as a COS.

Thank you for these comments. 

Reviewer #2: Thank you for asking me to review this submitted manuscript.

The authors report an international patient and healthcare professional consensus and propose a Cauda Equina Syndrome Core Outcome Set (CESCOS).

Some comments:

1. Overall, the authors seem to be appropriately building on their previous work [(Spine (Phila Pa 1976). 2018 Sep 1;43(17):E1005-E1013. doi: 10.1097/BRS.0000000000002605)], where significant heterogeneity in outcomes reported for studies after surgery for CES patients indicated a need for the development of a core outcome set.

2. Abstract: reasonable depiction of the manuscript.

3. Introduction: fine

4. Methods: described logically and comprehensively.

5. Results:

- In "Protocol Deviations", the first sentence is difficult to understand due to the word "out". Please rephrase.

This sentence has been changed from this:

“As mentioned before, the definition of consensus out was changed for the Delphi survey with agreement from the study team. There were no other deviations from the protocol.”

To this:

“As mentioned before, the definition for an outcome not to be included (termed “consensus out” in Table 1) was changed for the Delphi survey with agreement from the study team. There were no other deviations from the protocol.”

- The first two subheadings ('Protocol deviations' and 'The List of Outcomes') seem to be more appropriately placed in the Methods section.

- please reserve the Results section for Results alone.

Protocol deviations are to be reported in the results section in accordance with item 11 in Table 1 of the Core Outcome Set-STAndards for Reporting (COS-STAR)(1) guidelines.

The List of Outcomes has now been removed from the results and implemented into the paragraph titled “Information sources” in Methods as per your comments and in line with item 6a and 6b of the COS-STAR reporting guidelines. 

References

Kirkham JJ, Gorst S, Altman DG, Blazeby JM, Clarke M, Devane D, Gargon E, Moher D, Schmitt J, Tugwell P, Tunis S. Core outcome set–STAndards for reporting: the COS-STAR statement. PLoS medicine. 2016 Oct 18;13(10):e1002148.

- In reporting the "years of practice" amongst HCP participants, do the authors include years in training?

No. It is the years of practice as a consultant or attending surgeon after Board certification. This has now been highlighted in brackets within the Table 3 heading for clarification as “Years of practice (as a consultant/ attending after board certification).”

- Can the authors clarify how many respondents were trainees vs specialists?

All were specialists. No trainees were included. This was addressed as per the question above.

- Their later comment in the discussion that "63% had 10 years or more experience" is noted, but it remains unclear whether the 10 is post-residency training.

Yes 10 years’ post residency training. Sentence changed for clarification to:

“Most HCPs taking part in the Delphi were of a surgical background and 63% had 10 years or more experience after board certification as a consultant/ attending or the equivalent.”

6. Discussion:

- appropriate discussion of the consensus group composition.

This was added to the results section under “consensus meeting”:

“Of the 18 HCPs at the consensus meeting, 10 were surgeons involved in acute CES management and 8 were doctors and allied HCPs involved in the longer-term care and rehabilitation of CES patients. In the patient group, there was an equal spread of patients in the years since diagnosis of CES (<2: 5, ≥2<5: 4, ≥5<10: 6 and ≥10: 1).”

This was added in the discussion:

“As mentioned in the results, over half of the HCPs attending the consensus meeting were involved in acute management and the rest involved in longer term care and rehabilitation of CES patients, which would be reflective of a group of HCPs that manage CES patients in the short and long term from diagnosis. There was an equal spread of patients in the years since CES diagnosis, which would have also facilitated prioritisation of short and long term outcomes.”

- the authors have omitted to cite a recent study (perhaps because this was published after the present study's submission) which seemingly reported the largest cohort of patients with CES investigated for long-term outcomes using validated questionnaires. Please discuss the findings and choice of questionnaires used, and how your COS might improve this. [Hazelwood JE et al. An assessment of patient-reported long-term outcomes following surgery for cauda equina syndrome. Acta Neurochir (Wien). 2019 Sep;161(9):1887-1894. doi: 10.1007/s00701-019-03973-7. Epub 2019 Jul 1.]

Thank you for bringing us to the attention of this study. This has now been referenced in the discussion:

“A prospective study of long term outcomes after surgery for 46 CES patients had a mean follow up of 43 months. Validated questionnaires and unvalidated semi structured interviews were used to assess long term outcomes of bladder, bowel, sexual and physical function. Not all the outcomes in the CESCOS have been measured. For example, perineal sensation, sensation in genitals, leg muscle strength, pain due to abnormal sensation or non-painful stimulus and complications from the operation, were not measured. For the outcomes that were measured, there has not been a transparent consensus process(2) regarding the choice of these outcome measurement instruments. In Table 5 of a systematic literature review(3), it was shown that between studies, there is a lack of uniformity in the assessments used for the outcomes in CES, which makes it difficult to synthesise the results for meaningful analysis. The CESCOS highlights the outcomes for which this process must be undertaken in a transparent and methodologically sound manner.”

References

Prinsen CA, Vohra S, Rose MR, Boers M, Tugwell P, Clarke M, Williamson PR, Terwee CB. How to select outcome measurement instruments for outcomes included in a “Core Outcome Set”–a practical guideline. Trials. 2016 Dec;17(1):449.

Srikandarajah N, Wilby M, Clark S, Noble A, Williamson P, Marson T. Outcomes reported after surgery for cauda equina syndrome: a systematic literature review. Spine. 2018 Sep 1;43(17):E1005.

- Limitations: fair discussion.

- can the authors discuss the potential uptake of their recommended COS, incl the length of time required to obtain the COS data and any pragmatic difficulties.

This has been added to the discussion:

There is little research into the uptake of core outcome sets in comparison to randomised trials and systematic reviews as there are a relatively smaller number of them(4). A review of the rheumatoid arthritis COS established in 1994 showed that 81% of trials between 2002 and 2016 were reporting it (5). Implementation is an important aspect to help promote uptake and to aid this the CESCOS is registered on the COMET database, is published and will be presented at meetings and relevant HCP, patient and research funding bodies will be informed. 

In the qualitative interviews, Delphi survey and consensus meeting for the CESCOS study, patients were keen and willing to be involved. This may be a reflection that it is a rare syndrome so any attention or further research for the condition is actively engaged and welcomed by them suggesting the burden of data collection is not an issue with this patient group. A study reviewing the long-term outcomes after spinal surgery for CES showed that there was a real need for HCPs to spend sufficient time discussing the difficult issues and delivering prognostic information to patients regarding their outcomes(6). This highlights the importance of recording and reporting the CESCOS outcomes for future research studies. Currently, a multicentre prospective observational cohort study in CES is using the CESCOS(7) for its follow up data collection. The pragmatic difficulties of data measurement will be explored here. The COS should be reviewed in the future to see if any outcomes need to be added or subtracted(19). The aim is to do this in five years to analyse uptake in CES research studies. 

References

Kirkham JJ, Clarke M, Williamson PR. A methodological approach for assessing the uptake of core outcome sets using ClinicalTrials. gov: findings from a review of randomised controlled trials of rheumatoid arthritis. bmj. 2017 May 17;357:j2262.

Woodfield J, Hoeritzauer I, Jamjoom AA, Pronin S, Srikandarajah N, Poon M, Roy H, Demetriades AK, Sell P, Eames N, Statham PF. Understanding cauda equina syndrome: protocol for a UK multicentre prospective observational cohort study. BMJ open. 2018 Dec 1;8(12):e025230.

Korse NS, Veldman AB, Peul WC, Vleggeert-Lankamp CL. The long term outcome of micturition, defecation and sexual function after spinal surgery for cauda equina syndrome. PLoS ONE. 2017;12(4):e0175987.

7. Conclusion:

- 16 outcomes that are critical to key stakeholders have been identified. How

How this has been identified is explained in the methods and the results of this paper. There is also a protocol which we have referenced in the manuscript. 

8. This constitutes a piece of hard work and the authors are congratulated in its design and execution. Looking forward to a quick return addressing the points made.

Thank you for these comments. I hope the answers and corrections made are satisfactory. 

1. Kirkham JJ, Gorst S, Altman DG, Blazeby JM, Clarke M, Devane D, et al. Core outcome set–STAndards for reporting: the COS-STAR statement. PLoS medicine. 2016;13(10):e1002148.

2. Prinsen CA, Vohra S, Rose MR, Boers M, Tugwell P, Clarke M, et al. How to select outcome measurement instruments for outcomes included in a “Core Outcome Set”–a practical guideline. Trials. 2016;17(1):449.

3. Srikandarajah N, Wilby M, Clark S, Noble A, Williamson P, Marson T. Outcomes reported after surgery for Cauda Equina Syndrome: A Systematic Literature Review. Spine. 2018.

4. Williamson PR, Altman DG, Bagley H, Barnes KL, Blazeby JM, Brookes ST, et al. The COMET handbook: version 1.0. Trials. 2017;18(3):280.

5. Kirkham JJ, Clarke M, Williamson PR. A methodological approach for assessing the uptake of core outcome sets using ClinicalTrials. gov: findings from a review of randomised controlled trials of rheumatoid arthritis. Bmj. 2017;357:j2262.

6. Korse NS, Veldman AB, Peul WC, Vleggeert-Lankamp CL. The long term outcome of micturition, defecation and sexual function after spinal surgery for cauda equina syndrome. PLoS ONE. 2017;12(4):e0175987.

7. Woodfield J, Hoeritzauer I, Jamjoom AA, Pronin S, Srikandarajah N, Poon M, et al. Understanding cauda equina syndrome: protocol for a UK multicentre prospective observational cohort study. BMJ open. 2018;8(12):e025230.

---

## [Editor Report · Decision Letter 1]

15 Nov 2019

Cauda Equina Syndrome Core Outcome Set (CESCOS): An international patient and healthcare professional consensus for research studies

PONE-D-19-18530R1

Dear Dr. Srikandarajah,

We are pleased to inform you that your manuscript has been judged scientifically suitable for publication and will be formally accepted for publication once it complies with all outstanding technical requirements.

With kind regards,

Rodrigo Romao

Academic Editor

PLOS ONE

Additional Editor Comments (optional):

The authors have adequately addressed the questions and comments posed by the reviewers.
---

## [Editor Report · Acceptance letter]

18 Dec 2019

PONE-D-19-18530R1 

Cauda Equina Syndrome Core Outcome Set (CESCOS): An international patient and healthcare professional consensus for research studies 

Dear Dr. Srikandarajah:

I am pleased to inform you that your manuscript has been deemed suitable for publication in PLOS ONE. Congratulations! Your manuscript is now with our production department. 

With kind regards,

on behalf of

Dr. Rodrigo Romao 

Academic Editor

PLOS ONE